# Patients with Systemic Sclerosis with and without Overlap Syndrome Show Similar Microvascular Abnormalities

**DOI:** 10.3390/diagnostics11091606

**Published:** 2021-09-03

**Authors:** Gabriella Nagy, László Czirják, Gábor Kumánovics

**Affiliations:** Department of Rheumatology and Immunology, Medical School, University of Pécs, Akác Street 1, 7362 Pécs, Hungary; nagy.gabriella@pte.hu (G.N.); czirjak.laszlo@pte.hu (L.C.)

**Keywords:** SSc pattern, capillary density, nailfold video capillaroscopy, connective tissue diseases, systemic sclerosis

## Abstract

Introduction: Nailfold video capillaroscopy (NVC) is a useful tool for measuring capillary density (CD) and capillary morphology parameters and is mainly used in systemic sclerosis (SSc). Objective: We aimed to assess the prevalence of an SSc pattern and CD in different connective tissue diseases (CTDs). Methods: NVC was performed on 268 patients with CTDs. Control groups consisted of 104 healthy volunteers (HVs) and 36 primary Raynaud’s patients (PRPs). Results: Decreased CD was more prevalent in SSc, systemic lupus erythematosus (SLE), inflammatory myopathies (IIM), and overlap CTD patients compared with both controls. Average CD, the prevalence of decreased CD, and the prevalence of an SSc pattern did not differ significantly between SSc patients with (*n* = 39) and without (*n* = 50) overlap syndrome. An SSc pattern was significantly more prevalent in SLE (23%), SSc (82%), IIM (35%), and rheumatoid arthritis (17%) compared with both control groups. The prevalence of an elevated microangiopathy evaluation score (MES) was significantly higher in SLE, SSc, and IIM than in the HVs. Conclusion: The presence of another CTD in SSc did not influence CD or morphology. An SSc pattern may also be present in CTDs other than SSc. The MES is a useful instrument to distinguish between patients with CTDs and controls.

## 1. Introduction

Capillaroscopy is a non-invasive method used to evaluate capillary density (CD) and morphology. This method has an important role in differentiating between primary and secondary Raynaud’s phenomenon (RP) [1,2,3] and a key role in the recognition of cases with very early systemic sclerosis (SSc) [3]. In recent decades, several methods, including nailfold videocapillaroscopy (NVC), have been introduced with a standardized nomenclature of microvascular abnormalities [4]. Recently, a high-quality guideline on microvascular investigations was published, including a description of patient preparation, the device, and the examination to ease comparison with studies on NVC, especially in cases other than SSc [5].

Capillaroscopic abnormalities are widely described in SSc [1,3,6,7]. Capillary density is one of the most important parameters in SSc [8,9]; decreased density correlates with the severity of internal organ involvement or anticipates a new digital ulcer development [9,10,11,12,13,14,15]. Based on the extent of SSc-associated NVC changes, three different patterns were identified [6]. An early SSc pattern is characterized by a few enlarged/giant loops, hemorrhages, and a preserved CD. An active pattern shows more severe changes, including a higher prevalence of giant loops and hemorrhages, a moderate loss of CD, and proof of revascularization. A late pattern is characterized by the most disorganized morphological abnormalities, including severe capillary loss with the domination of ramified loops [6]. To simplify the differentiation between an SSc pattern and no SSc pattern, a fast-track algorithm was recently developed [16]. This particular recommendation includes simple rules: if the CD is a minimum of 7 mm and no giant capillaries are present, an SSc pattern is not confirmed, but if the CD is a maximum of 3 mm with abnormal shapes, an SSc pattern can be declared. If the picture does not match any of the rules, an SSc pattern cannot be stated [16]. Similar to this algorithm, the cut-off for normal CD is 7 per mm in healthy subjects [17].

In addition to the assessment of CD and an SSc pattern, other complex methods have been developed to describe microangiopathy in SSc [18]. The microangiopathy evaluation score (MES) enables a semi-quantitative evaluation of three different parameters scored in a semi-quantitative way, including the loss of capillaries, disorganization of the microvascular array, and capillary ramifications [18]. In the same semi-quantitative scale, other SSc-related capillaroscopic parameters may be assessed, including hemorrhages, dilatation, and giant capillaries. The NVC parameters, including density, capillary dimensions, and hemorrhages but not ramifications, showed good interobserver reliability; therefore, simplified capillaroscopic definitions were defined [19]. The use of a simple normal (hairpin, crossing, or tortuous) and abnormal (not hairpin, crossing, or tortuous) classification has been recommended recently [4] to minimalize difficulties in data presentation and comparison. Due to the complexity of the calculation of complex SSc-related scores, these are mainly used for scientific purposes.

Concerning systemic lupus erythematosus (SLE), idiopathic inflammatory myopathies (IIM,) and Sjögren’s syndrome (SS), systematic reviews on microvascular abnormalities have been published recently [20,21,22,23,24]. In SLE, non-specific alterations were mainly described against healthy volunteers (HVs), and a significantly lower capillary density was reported in SLE patients with RP [20]. With the exception of a few cohorts [24], the majority of the studies on the NVC changes of SLE were conducted in small samples [20], and an SSc pattern was only rarely observed (2.4–15%) [23,24,25].

Capillary enlargement and/or giant capillaries are often reported in patients with adult IIM [21,26,27]. Based on the systematic review by Bertolazzi, rapid changes in the microvascular picture can be observed in IIM with the early appearance of signs of revascularization [21]. An SSc pattern was reported in 26.9–88.9% [21] of myositis patients.

A systematic analysis of the few available SS-associated capillaroscopic publications revealed a lower CD compared to HVs. An SSc pattern may also be present in SS, mostly in cases with overlap syndrome or overlapping features, and a higher prevalence of an SSc pattern was reported in RP-associated cases (25%). The review suggests that NVC performed on patients with SS may identify those with a higher risk of developing overlap syndrome (e.g., SSc) and subclassify patients at a higher risk of severe disease [22].

A few NVC studies are available on other connective tissue diseases (CTDs), including rheumatoid arthritis (RA) and antiphospholipid syndrome (APS). APS is characterized by widened afferent, apical, and efferent diameters [28]. Previously in RA, elongated loops and the presence of prominent subpapillary veins were described [29]. Dilated capillaries were also reported to appear more frequently in patients with RP [29]. Scleroderma capillaroscopic pattern was reported in 0.5–20.9% of patients with RA [26,29,30].

In our study, we aimed to investigate the prevalence of a decreased CD and an SSc pattern in different patients with CTD in a standardized way in a single tertiary care center and compare the possible differences of microvasculature between patients with systemic sclerosis with and without overlap syndromes. In addition to the commonly used parameters (CD and SSc pattern), we also aimed to assess the possible usefulness of the more accurate but complex technique, the MES.

## 2. Materials and Methods

### 2.1. Patient Selection

Two hundred and sixty-eight patients with CTDs and 104 healthy volunteers, as well as 36 cases with primary Raynaud’s phenomenon (PRP), were investigated in our tertiary care center between 2015 and 2018. Regarding RP associated with SSc, SLE, RA, and SS cases, every second consecutive patient was included in the study. Each consecutive patient with SLE, RA, and SS who did not have RP was also enrolled. All consecutive patients with IIM, APS, and vasculitis attending our tertiary care center were enrolled. All patients were interviewed and examined by the same investigator (GN) using a standard protocol regarding key symptoms and complaints of CTDs independent of the already established clinical diagnosis of the particular patient [31,32,33,34,35,36,37,38,39]. Patients with overlap syndrome were defined as patients fulfilling at least to classification criteria [31,32,33,34,35,36,37,38,39]. Both treatment naïve and treated patients were enrolled in our study.

As controls, 104 HVs without a previous history of CTD, RP, hypertension, diabetes mellitus, or drug abuse and 36 patients with PRP were enrolled.

All enrolled patients completed a self-assessment questionnaire and were interviewed about the presence and distribution of RP based on the 2014 preliminary criteria of Raynaud’s syndrome [40]. The interview regarding the presence of RP and investigation of the fulfillment of the classification criteria of SSc, SLE, RA, APS, SS, IIM, and ANCA vasculitis were performed by the same investigator for each patient (GN).

### 2.2. Laboratory Investigations

Antinuclear antibodies were tested with Quanta Lite ANA ELISA kit (Inova Diagnostics, Ref 708750, San Diego, CA, USA), and for the assessment of anti-topoisomerase I (anti-Scl70) antibodies, ORG 514 ELISA assay (Orgentec, Mainz, Germany) was used. Anticentromer antibodies were tested with ORG 633 ELISA Assay (Orgentec, Mainz, Germany), and anti-RNA polymerase III was evaluated with Euroimmun DL 15321601 immunoblot assay (Mountain Lake, VA, USA).

### 2.3. Capillaroscopic Investigation

An NVC investigation with a magnification of 200× was performed with a drop of paraffin oil and was recorded and evaluated by the same investigator (GN) (Videocap, DS Medica, Milano, Italy). Before performing the NVC examination, all subjects remained at room temperature for 15–20 min. The NVC was performed on eight fingers on both hands except for the thumbs. Four 1 mm length areas on all investigated fingers were assessed. The investigated parameters included capillary density per millimeter. The presence of an SSc pattern was assessed, and in the case of a presence, it was subcategorized to an early, active, or late SSc pattern [6]. The MES was also calculated [18].

As the normal capillary density was defined as 7 mm based on previous papers [17], patients were subdivided into patients with a normal average density (≥7 mm) and patients with a decreased density (<7 mm).

### 2.4. Statistical Analysis

Based on previous publications, an a priori sample size calculation with a power of 80% and α = 0.05 was performed to best differentiate CD between HVs and patients with different CTDs [28,41,42,43,44]. The number of patients who needed to be examined with a different diagnosis was 19 for SSc, 103 for SLE, 25 for IIM, 31 for SS, 402 for APS, and 15 for RA. Normality was tested with the Kolmogorov–Smirnov test and, according to the distribution *t*-test, a Kruskal–Wallis, an ANOVA, or a Mann–Whitney U test was used to compare the data. Nonparametric variables were tested with Fisher’s exact test or a chi-squared test as appropriate. As most of the data showed a non-normal distribution, the results are shown as median (lower quartile; upper quartile) values. For the statistical analysis, STATISTICA version 6.0 (Stat Soft Inc., 2001, Tulsa, OK, USA) was used.

Receiver operating characteristic (ROC) curves were used to examine the performance of the MES and CD to differentiate between patients with CTDs and the controls (HVs and PRPs). For these calculations, StatPlus AnalystSoft Inc. version 7 (AnalystSoft Inc., 2018, Walnut, CA, USA) was used.

## 3. Results

### 3.1. Clinical Characteristics

The main clinical characteristics of the investigated patient groups and the controls are summarized in Table 1.

Out of the 39 patients with overlap SSc, 29 patients fulfilled two classification criteria. SSc-SLE overlap was present in six cases, SSc-APS overlap was detected in two cases, SSc-IIM in five cases, SSc-SS overlap in nine cases, RA overlap in six cases, and in the case of one patient, ANCA-associated vasculitis was classified in addition to SSc. In the case of eight patients, three different classification criteria were fulfilled (SSc-SLE-APS *n* = 1, SSc-SLE-SS *n* = 3, SSc-SS-RA *n* = 2, SSc-SLE-RA *n* = 1, and SSc-IIM-SS *n* = 1, respectively). In the case of two patients, SSc-SLE-SS-RA was classified.

### 3.2. Assessment of Capillary Density

Patients with systemic sclerosis both with and without overlap syndrome had a significantly decreased CD compared with both the HVs and patients with PRP but not compared with each other (6.37/4.84–7.60/ vs. 6.97/5.77–8.33/, *p* = 0.062, respectively; Table 2 and Figure 1). Patients with SLE and IIM had a significantly decreased CD compared with both the HVs and patients with PRP. No significant difference regarding CD was observed between APS, SS, RA, and vasculitis patients and the HVs or patients with PRP. The prevalence of patients with a decreased CD did not differ significantly between patients with SSc with and without an overlap syndrome (56% vs. 51%, *p* = 0.674). The prevalence of a decreased CD was significantly higher in patients with SLE, IIM, and overlap CTD compared with the HVs and patients with PRP (Table 2). No significant difference between the HVs and patients with PRP was observed on either the CD median values or the prevalence of a decreased CD.

Patients with SSc positive for ANA had significantly higher CD compared with ones without ANA positivity (6.0/4.9; 7.5/ vs. 7.8/7.0; 8.4/ *p* < 0.001), and prevalence of an SSc pattern and elevated MES was also significantly elevated in these particular patients (88.5% vs. 46.4%, *p* < 0.001 and 86.9% vs. 60.7%, *p* < 0.001, respectively).

Patients with RP in the case of SLE and RA had significantly decreased CD compared with their counterparts without RP. No significant difference in SSc pattern was observed in comparison to patients with and without RP in patients with CTD (Table 3).

### 3.3. Qualitative Assessment of Microvascular Abnormalities

The prevalence of an SSc pattern was not significantly different in patients with SSc with and without overlap syndrome (67% vs. 82%, *p* = 0.137). The presence of SSc pattern was present in SLE in 23%, in APS in 18%, in IIM in 35%, in SS in 27%, and in RA in 17%. Patients with vasculitis showed no SSc pattern. The prevalence of an SSc pattern was significantly higher in SLE, APS, IIM, SS, and RA compared with both the HVs and patients with PRP (Table 2 and Figure 2).

CD was significantly lower in patients with an SSc pattern in the case of SLE, APS, and RA but not in IIM or SS (Table 4).

### 3.4. Assessment of Microvascular Abnormalities by Semi-Qualitative Methods

An area under the curve (AUC) analysis was used to estimate the optimal cut-off value of the MES. The value of 1.0625 was the best for differentiating patients with CTDs and controls (PRP + HV) (AUC, 0.786; sensitivity, 0.678; specificity, 0.735) and the value of 0.738 for differentiating between patients with SSc and the HVs (AUC, 0.953; sensitivity, 0.786; specificity, 0.951) (Figure 3).

Elevated MES showed the highest prevalence in patients with SSc without overlap syndrome. The prevalence of elevated MES was significantly higher in both SSc subgroups and the overlap CTD group compared with both control groups (HVs and patients with PRP) (Figure 4). A significant difference was only observed against HVs in the case of SLE, IIM, SS, and patients with PRP (Table 2).

## 4. Discussion

A large comparative study on capillary density and the prevalence of SSc patterns was performed on patients with CTD in a single tertiary care center.

In accordance with previous studies, a decreased CD was observed in our patients with SSc [1,6,7,10,18,45]. We raised the question of whether the presence of an overlap systemic autoimmune disease influences CD and morphology in SSc. We demonstrated that the prevalence of a decreased CD and the SSc pattern did not differ significantly in these particular subgroups, but a tendency was observed; patients in the SSc subgroup without overlap syndrome had a lower CD and a higher prevalence of a decreased CD than in those with overlap syndrome. Overlap syndromes might contribute to less severe microvascular changes in SSc. To the best of our knowledge, a systematic comparison between overlap and ‘pure’ SSc cases on CD has not been performed previously. We observed a 43.9% prevalence of SSc overlap syndrome as opposed to the previously observed 10–12% [46,47]. The possible explanation is that we used a protocol with a detailed search for every single overlap syndrome.

Vascular complications are more prevalent in ANA-positive patients with SSc [48,49], including pulmonary arterial hypertension and digital ulcers. These complications are generally associated with decreased CD [6,7,11,12]. Our results are in line with the previous observation, though a low prevalence of ANA-negative patients with SSc (69%) was observed, e.g., compared with the prevalence of the EUSTAR database (93.4%) [50]. This might be explained by the method of assessment.

Our results are in accordance with previous findings; a decreased CD was present in our patients with SLE compared with the HVs [20,21,24,26]. The SSc pattern showed a 23% prevalence in SLE; two-thirds of these particular cases showed SSc late pattern. This is a higher prevalence than previously reported (2.4–15%) [23,24,25], which may be explained by the high prevalence of RP among our cases with SLE, potentially due to the use of a standardized questionnaire for the assessment of RP. Otherwise, our cases with SLE do not differ substantially from other previously reported cohorts [20,23,24,25,26].

In the case of IIM, the median CD was significantly lower compared with the controls, but when the number of patients with a decreased CD (<7 mm) was compared with the controls, the significant difference disappeared. The prevalence of the SSc pattern in IIM is 35% in our cohort, which is within the previously observed range (26.9–88.9%) [21]. This mild difference might be explained by our cross-sectional study design, as microvascular changes might potentially change over time and treatment. An SSc pattern is highly variable in IIM, and it can rapidly change; therefore, timing is important in performing an NVC examination [21].

Patients with RA, SS, APS, and vasculitis did not show a significant difference compared with our controls either when the median values of CD or when the prevalence of a decreased CD were investigated. Our results on CD in the group of patients with SS are not in accordance with previous findings, as Capobianco et al. discovered a significantly higher CD compared with the HVs [51] and another research group also discovered a decreased CD compared with the HVs in SS [42]. In both investigations, CD was within a normal range. A small sample size might explain this difference between the literature and our cohort.

We demonstrated that the average CD is normal in both RA and APS, indicating a low level of microvascular damage in these diseases. Previous studies on RA and APS did not investigate CD. Our results show that a decrease in CD is not the hallmark of these diseases. In SS, APS, and RA, a significantly higher prevalence of the SSc pattern was observed compared with both the HVs and patients with PRP (Table 2), and the particular cases with APS or RA with the SSc pattern showed a significantly lower CD compared with the cases without the SSc pattern. Patients with SLE-SS and SLE-APS overlap also showed a decreased CD compared with the controls (Table 2). The number of investigated cases was low; further studies are required to clarify whether these particular patients with overlap CTD may belong to a special subgroup with more pronounced microvascular abnormalities.

A recent systematic review on vasculitis described the presence of SSc-specific abnormalities in GPA (a high prevalence of avascularity and microhemorrhages); however, CD was not investigated [52]. In our small ANCA-associated vasculitis cohort, we observed preserved CDs, indicating the absence of significant avascularity. Further investigations with a higher sample size are essential to draw conclusions on capillaroscopy abnormalities in ANCA-associated vasculitis.

In addition to the widely used parameters, we aimed to investigate the possible usefulness of a more complex method, the MES. The complex score was developed and used in research focusing solely on SSc. As it sums different SSc-related capillaroscopic parameters, it can more accurately reflect microvascular changes. As no standard cut-off is available regarding the MES, we performed a ROC analysis to detect patients with and without an abnormal MES. Our results suggest that MES values may differentiate between patients with SSc, SLE, and IIM and HVs. The MES can differentiate between the HVs and PRPs. As the frequently used methods, including CD and the presence of an SSc pattern, cannot discriminate between these particular groups of patients, the MES clearly has additional value. Further validation steps are essential to clarify the role of the MES in CTDs other than SSc.

The strength of our study is that a high number of patients was evaluated by the same investigator with the same standard protocol, including clinical laboratory findings, classification, and capillaroscopic evaluation. The presence of Raynaud’s phenomenon and overlap syndrome was carefully assessed. The weakness of the study is that this is a single tertiary center study, which may cause bias in patient selection. Future multicenter studies are required with a standardized methodology and patient selection to clarify the importance of our current findings. A limitation of our study is that we included patients with hypertension and diabetes, and these particular comorbidities might have an impact on our capillaroscopy findings [53,54]. Another weakness of our study is that the disease activity and treatment were not evaluated in different CTDs.

Our overall assumption is that a decreased CD and the prevalence of an SSc pattern are similar in pure SSc and overlap SSc. These capillaroscopic changes, including the prevalence of an SSc pattern and decreased CD, are more prevalent than previously suggested in CTDs other than SSc, including patients with SLE, IIM, and overlap syndromes. A more extensive method, the MES could also accurately distinguish between patients with CTDs and controls even between HVs and patients with PRP besides the widely used CD.

In summary, a capillaroscopic examination should include both the evaluation of capillary density and an assessment of the SSc pattern. The complex MES score may be a promising candidate to be studied in future follow-up studies assessing microvascular changes.

## Figures and Tables

**Figure 1 diagnostics-11-01606-f001:**
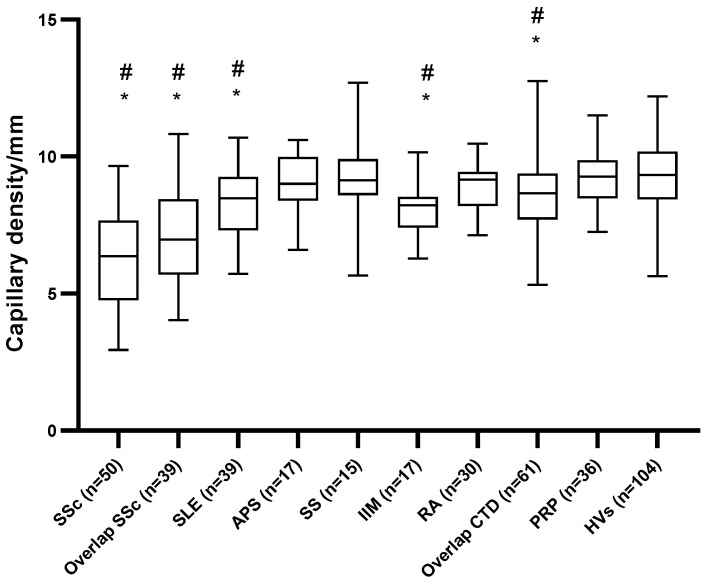
Capillary density in the investigated groups of patients. * *p* < 0.05 or lower significance compared with PRP. ^#^
*p* < 0.05 or lower significance compared with HV. SSc, systemic sclerosis; SLE, systemic lupus erythematosus; APS, antiphospholipid syndrome; SS, Sjögren’s syndrome; IIM, idiopathic inflammatory myositis; RA, rheumatoid arthritis; CTD, connective tissue disease; PRP, primary Raynaud’s syndrome; HVs: healthy volunteers.

**Figure 2 diagnostics-11-01606-f002:**
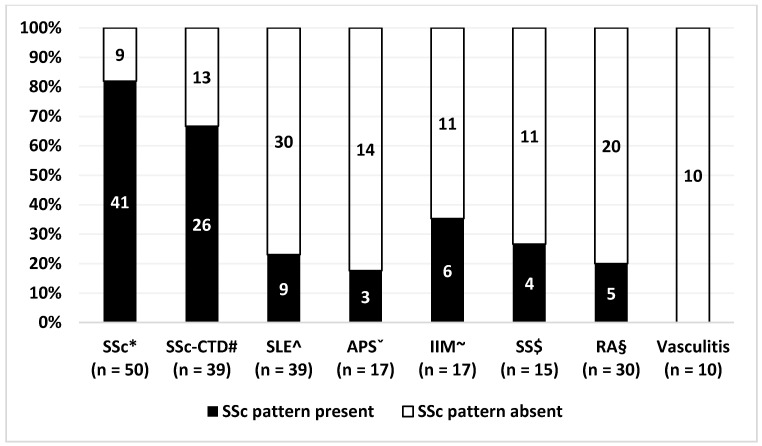
Prevalence of SSc pattern in different connective tissue disease groups: * SSc, systemic sclerosis; # SSc-CTD, systemic sclerosis overlap syndrome; ^ SLE, systemic lupus erythematosus; ˇ APS, antiphospholipid syndrome; ~ IIM, idiopathic inflammatory myositis; $ SS, Sjögren’s syndrome; § RA, rheumatoid arthritis.

**Figure 3 diagnostics-11-01606-f003:**
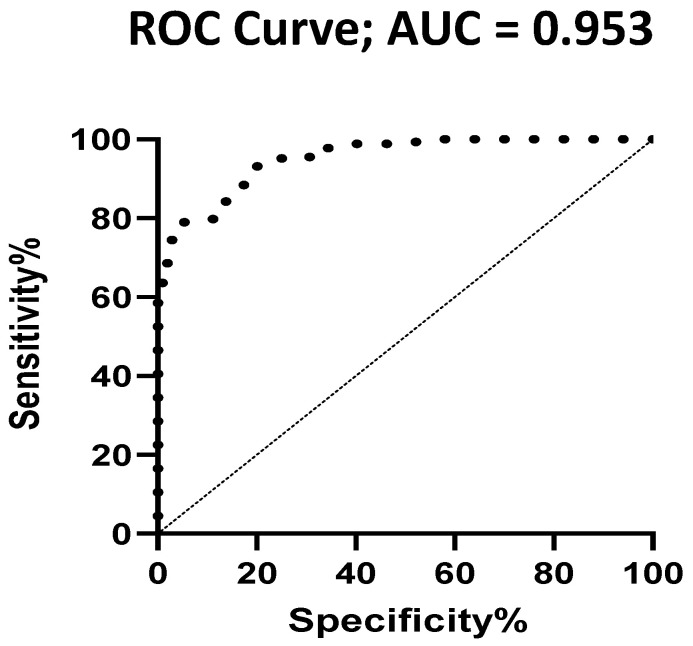
ROC curves and AUC for estimating the best differentiating accuracy of the microangiopathy evaluation score.

**Figure 4 diagnostics-11-01606-f004:**
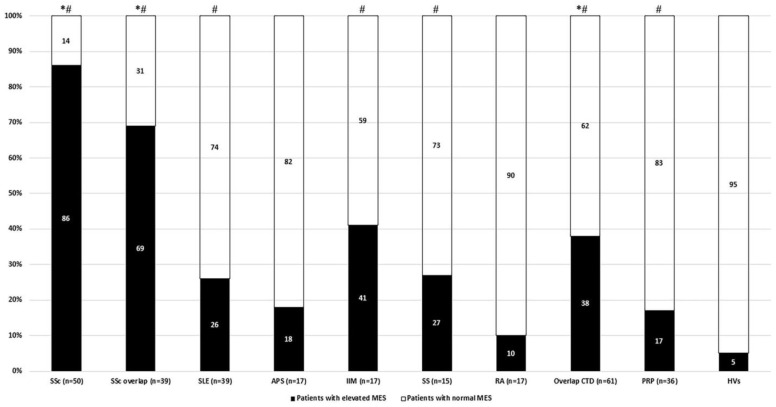
Prevalence of elevated MES in the investigated groups of patients. * *p* < 0.05 or lower significance compared with PRP. ^#^
*p* < 0.05 or lower significance compared with HV.

**Table 1 diagnostics-11-01606-t001:** Clinical and laboratory findings of 418 investigated subjects.

	SSc ^b^			RA ^j^	SLE ^k^	APS ^l^	IIM ^m^	SS ^n^	Overlap CTD Patients ^c^	PRP ^d^	HV ^e^
	All	Without Overlap	With Overlap								
Number of patients	89	50	39	30	39	17	17	15	61	36	104
Mean age (year) ^a^	58 (52–65)	58 (52–65)	59 (51–64)	58 (44–62)	50 (40–57)	60 (55–63)	59 (49–62)	52 (49–66)	57 (51–64)	47 * (32–54)	51 * (38–61)
Number of females (%)	79 (89)	41 (82)	38 (97) *	26 (87)	32 (82)	15 (88)	11 (65)	13 (87)	56 (91)	35 (97) *	80 (77)
Disease duration (year) ^a^	9 (4–17)	6 (3–15)	15 * (6–18)	8 (5–11)	10 (6–15)	11 (7–17)	4 (1–9)	6 (5–11)	12 (7–19)	3 * (0.75–11)	–
Number of patients (%) with											
ANA ^f^	61 (69)	33 (66)	28 (72)	3 (10) *	35 (90) *	0 (0) *	5 (29) *	6 (40)	53 (87)	0 (0)	1 (1) *
Scl70 ^g^	16 (18)	11 (22)	5 (13)	0 (0) *	0 (0) *	0 (0) *	0 (0) *	0 (0) *	2 (3) *	0 (0)	0 (0) *
RNAPIII ^h^	2 (2)	2 (4)	0 (0)	0 (0)	4 (10)	0 (0)	0 (0)	0 (0)	2 (3)	0 (0)	0 (0)
ACA ^i^	27 (30)	18 (36)	9 (23)	0 (0) *	0 (0) *	0 (0) *	0 (0) *	0 (0) *	3 (5) *	0 (0)	1 (1) *
Number of patients with Raynaud’s syndrome (%)	88 (99)	50 (100)	38 (97)	12 (40) *	24 (62) *	14 (82) *	5 (29) *	12 (80) *	42 (69) *	36 (100)	0 (0) *
Number of patients with current smoking (%)	28 (31)	17 (34)	11 (28)	8 (27)	14 (36)	10 (59)	4 (24)	1 (7) *	17 (28)	10 (28)	0 (0) *
Number of patients with diabetes mellitus (%)	6 (7)	4 (8)	2 (5)	3 (10)	1 (3)	5 (29) *	2 (12)	2 (13)	2 (3)	1 (3)	0 (0) *
Number of patients with hypertension	44 (49)	24 (48)	20 (0)	13 (43)	22 (56)	10 (58)	5 (29)	8 (53)	49 (80) *	0 (0) *	0 (0) *

^a^ Results are presented as median values (25th–75th percentile); ^b^ SSc, systemic sclerosis; ^c^ CTDs, connective tissue diseases; ^d^ PRP, primary Raynaud’s syndrome; ^e^ HV, healthy volunteers; ^f^ ANA, antinuclear antibody; ^g^ Scl70, anti-topoisomerase antibody; ^h^ RNAPIII, anti-RNA-polymerase-III antibody; ^i^ ACA, anti-centromere antibody; ^j^ RA, rheumatoid arthritis; ^k^ SLE, systemic lupus erythematosus; ^l^ APS, antiphospholipid syndrome; ^m^ IIM, idiopathic inflammatory myositis; ^n^ SS, Sjögren’s syndrome; * *p* < 0.05 compared with patients with SSc patients without overlap.

**Table 2 diagnostics-11-01606-t002:** Capillaroscopic parameters in the different groups of patients.

	Capillary Density (number/mm) ^a^	Number of Patients with Decreased (<7 mm) Density (%)	Number of Patients with SSc Pattern Early/Active/Late/All (%)			Number of Patients with Elevated MES (>1.0625) (%)
SSc ^b^ without overlap (*n* = 50)	6.37 *^#^ (4.84–7.60)	28 (56) *^#^	3	9	29	43 (86) *^#^
			41 (82) *^#^			
Overlap SSc (*n* = 39)	6.97 *^#^ (5.77–8.33)	20 (51) *^#^	2	1	23	27 (69) *^#^
			26 (67) *^#^			
SLE ^d^ (*n* = 39)	8.48 *^#^ (7.34–9.24)	7 (18) *^#^	3	0	6	10 (26) ^#^
			9 (23) *^#^			
APS ^e^ (*n* = 17)	9.00 (8.47–9.78)	1 (6)	0	0	3	3 (18)
			3 (18) *^#^			
IIM ^f^ (*n* = 17)	8.22 *^#^ (7.59–8.47)	2 (12)	1	0	5	7 (41) ^#^
			6 (35) *^#^			
SS ^g^ (*n* = 15)	9.13 (8.72–9.75)	2 (13)	2	0	2	4 (27) ^#^
			4 (27) *^#^			
RA ^h^ (*n* = 30)	9.16 (8.27–9.42)	0 (0)	1	2	2	3 (10)
			5 (17) *^#^			
Vasculitis ^i^ (*n* = 10)	8.71 (8.17–9.32)	1 (10)	0	0	0	1 (10)
			0 (0)			
CTD ^c^ with different overlaps (*n* = 61)	8.66 *^#^ (7.72–9.31)	7 (11) *^#^	4	1	6	23 (38) *^#^
			11 (18) *^#^			
PRP ^j^ (*n* = 36)	9.27 (8.55–9.85)	0 (0)	0	0	0	6 (17) ^#^
			0 (0)			
HV ^k^ (*n* = 104)	9.33 (8.46–10.16)	2 (2)	2	1	0	5 (5)
			3 (3)			

^a^ Results are represented as median values (percentile 25–75); ^b^ SSc, systemic sclerosis; ^c^ CTDs, connective tissue diseases; ^d^ SLE, systemic lupus erythematosus; ^e^ APS, antiphospholipid syndrome; ^f^ IIM, idiopathic inflammatory myopathies; ^g^ SS, Sjögren’s syndrome; ^h^ RA, rheumatoid arthritis; ^i^ vasculitis, patients with ANCA-associated vasculitis; ^j^ PRP, primary Raynaud’s phenomenon; ^k^ HV, healthy volunteers; * *p* < 0.05 or lower significance compared with PRP ^j^; ^#^
*p* < 0.05 or lower significance compared with HV ^k^.

**Table 3 diagnostics-11-01606-t003:** Comparison of capillary density and SSc pattern between patients with and without Raynaud’s phenomenon in patients with different connective tissue diseases.

		Patients with Raynaud’s Phenomenon	Patients Without Raynaud’s Phenomenon	*p* (Mann–Whitney U Test or Fisher’s Exact Test)
Systemic lupus erythematosus *n* = 39 (RP ^#^: *n* = 24; non-RP ^§^: *n* = 15)	Capillary density	7.9 (7.0;8.7)	9.3 (8.5;10.2)	<0.001
	SSc ^§^ pattern (*n*)	8	1	NS *
Antiphospholipid syndrome *n* = 17 (RP ^#^: *n* = 14; non-RP ^§^: *n* = 3)	Capillary density	9.0 (8.5;9.7)	9.8 (7.9;10.6)	NS *
	SSc ^§^ pattern (*n*)	3	0	NS *
Idiopathic inflammatory myositis *n* = 17 (RP ^#^: *n* = 5; non-RP ^§^: *n* = 12)	Capillary density	7.7 (6.9;8.2)	8.3 (7.7;8.6)	NS *
	SSc ^§^ pattern (*n*)	2	4	NS *
Sjögren’s syndrome *n* = 15 (RP ^#^: *n* = 12; non-RP ^§^: *n* = 3)	Capillary density	9.3 (8.8;10.0)	8.2 (6.0;9.43)	NS *
	SSc ^§^ pattern (*n*)	2	2	NS *
Rheumatoid arthritis *n* = 30 (RP ^#^: *n* = 12; non-RP ^§^: *n* = 18)	Capillary density	8.3 (7.7;9.1)	9.3 (9.0;9.8)	<0.01
	SSc ^§^ pattern (*n*)	3	2	NS *
ANCA-associated vasculitis *n* = 10 (RP ^#^: *n* = 2; non-RP ^§^: *n* = 8)	Capillary density	8.9 (8.5;9.3)	8.6 (8.2;9.5)	NS *
	SSc ^§^ pattern (*n*)	0	0	NA ^$^
Overlap connective tissue disease *n* = 61 (RP ^#^: *n* = 42; non-RP ^§^: *n* = 19)	Capillary density	8.4 (7.6;9.4)	8.7 (7.9;9.2)	NS *
	SSc ^§^ pattern (*n*)	8	3	NS *

* NS, not significant; ^$^ NA, non-applicable; ^§^ SSc, systemic sclerosis; ^#^ RP, patients with Raynaud’s phenomenon; ^§^ non-RP, patients without Raynaud’s phenomenon.

**Table 4 diagnostics-11-01606-t004:** Evaluation of capillary density in patients with and without SSc capillaroscopic pattern.

	Patients with SSc ^a^ Capillaroscopic Pattern /n/	Patients Without SSc a Capillaroscopic Pattern /n/	*p* (Mann–Whitney U Test)
Systemic lupus erythematosus	7.01 (6.8;7.81) /9/	8.69 (8.00;9.34) /30/	0.0017
Antiphospholipid syndrome	7.06 (6.59;8.1) /3/	9.38 (8.56;10.18) /14/	0.0196
Rheumatoid arthritis	7.09 (7.32;8.56) /5/	9.05 (8.69;9.50) /25/	0.0451
Idiopathic inflammatory myositis	7.07 (6.91;8.22) /6/	8.33 (7.66;8.63) /11/	0.0786
Sjögren’s syndrome	7.44 (5.84;8.04) /4/	11.88 (8.84;10.06) /11/	0.1172

^a^ SSc, systemic sclerosis.

## Data Availability

The data are not publicly available due to ethical issues.

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
