# Peer review of "Patients with Systemic Sclerosis with and without Overlap Syndrome Show Similar Microvascular Abnormalities"

_diagnostics, 2021, doi:10.3390/diagnostics11091606_

Round 1
Reviewer 1 Report
I have read the manuscript by Nagy et al. with great interest. The authors measured NVC parameters in patients with various CTDs. They concluded that these are similar in patients with SSc with and without overlap syndrome and different from healthy controls.
Major Comments:
- A major critical issue of this manuscript that the primary aim is not clear. The title suggests that that is to compare patients with SSc with and without overlap syndrome. In contrast, the final sentence of the Introduction says that to assess CD and SSc parameters in different CTDs. It is vital to assess the main message of the manuscript. Please, revise the manuscript and set a clear aim.
- In line with this, it is not clear for what primary objective the power analysis was based on.
- I understand that diabetes and hypertension were excluded in controls. But what about the patient group? Could inclusion of these comorbidities influence the results?
- Please, define “overlap syndrome”.
- Please clarify if they were treatment naïve.
- Table 1. Please perform comparisons between the groups. As mentioned above, please provide data on the relevant comorbidities.
Minor Comments:
- Please change the title to “Patients with Systemic Sclerosis with and without Overlap Syndrome…”
- Please define what you meant by overlap syndrome.
- Line 17. Please change “both controls” to “both control groups”
- Please add actual results in numbers. It is hard to evaluate the conclusions without seeing the figures.
- Throughout the manuscript it is inappropriate to use “CTD patients” or “SSc patients”. Please change these to patients with CTD or patients with SSc. For example, Line 74. Please, change to “patients with myositis”. Line 88. Please change to “patients with different CTDs”.
Author Response
Dear Reviewer ,
Thank for reviewing our manuscript. Regarding your comments, please find our answers below.
Major Comments
- A major critical issue of this manuscript that the primary aim is not clear. The title suggests that that is to compare patients with SSc with and without overlap syndrome. In contrast, the final sentence of the Introduction says that to assess CD and SSc parameters in different CTDs. It is vital to assess the main message of the manuscript. Please, revise the manuscript and set a clear aim.
We clarified the primary aim of this particular study as follows:
In our study, we aimed to investigate the prevalence of a decreased CD and SSc pattern in different CTD patients in a standardized way in a single tertiary care center and compare the possible differences of microvasculature between systemic sclerosis patients with and without overlap syndromes. In addition to the commonly used parameters (CD and SSc pattern), we also aimed to assess the possible usefulness of the more accurate but complex technique, the MES.
- In line with this, it is not clear for what primary objective the power analysis was based on.
A priori sample size calculation was based on capillary density, to best differentiate between different CTDs and healthy volunteers.
- I understand that diabetes and hypertension were excluded in controls. But what about the patient group? Could inclusion of these comorbidities influence the results?
Patients with different comorbidities including diabetes mellitus or hypertension were not excluded out from our assessment. In fact these particular comorbidities may influence our findings. We considered that these particular comorbidities may be present up to 25-30% among our patients. We did not considered to exclude such a significant number of patient from this study. In fact capillary rarefaction was described in these comorbidities (doi: 10.2152/jmi.68.6. and doi: 10.1016/j.mvr.2020.104036.), hence the inclusion of CTD patients with diabetes mellitus or hypertension possible might had influence on our results. As a limitation of the study, we included a remark in the discussion.
- Please, define “overlap syndrome”.
Overlap syndrome was defined as fulfilling at least two different CTD classification criteria [31][32][33][34][35][36][37][38][39],
- Please clarify if they were treatment naïve.
Treatment of enrolled patients were not taken into consideration, both treatment naïve and treated patients were enrolled in our study. Furthermore the mean follow up time was long making difficult to evaluate the effect of the different treatment regimens.
- Table 1. Please perform comparisons between the groups. As mentioned above, please provide data on the relevant comorbidities.
Table 1 was modified based on your recommendations, comparison between patient groups was added in addition to comorbidities and overlap CTD groups.
Minor comments
Thank you for your suggestion. We corrected them based on your recommendation.
Hopefully our answers achieve your standards and You will find our manuscript valuable for publication.
Yours sincerely.
Gábor Kumánovics MD, PhD
Reviewer 2 Report
The authors presented the results of videocapillaroscopic image analysis in patients with CTD diagnosed in a tertiary reference center. In addition to the standard CD and SSC pattern determination, the MES assessment was used. Due to the large number of patients, the work seems to be very interesting, but some issues require clarification before possible publication.
- Table 1 - The ratio of women to men in each subgroup should be presented as a percentage for easier comparison
- The percentage of patients with SSc and ANA negative is very high. The reason for this difference with the data on serum negative Ssc patients available in the literature should be discussed. What method was used to assess the presence of ANA and individual classes of antibodies?
- All CTD patients were included in the analysis. What were the CD and SSc-pattern data when only patients with Raynaud's phenomenon were considered?
- Characteristics of "SSc overlap" and "CTD overlap" groups are missing
- The conclusions are too general
Author Response
Dear Reviewer,
Thank for reviewing our manuscript. Regarding your comments, please find our answers below.
Comments
- Table 1 - The ratio of women to men in each subgroup should be presented as a percentage for easier comparison
Thanks for your advice, we have corrected it.
- The percentage of patients with SSc and ANA negative is very high. The reason for this difference with the data on serum negative Ssc patients available in the literature should be discussed. What method was used to assess the presence of ANA and individual classes of antibodies?
Antinuclear antibodies were tested with Quanta Lite ANA ELISA kit, (Ref 708750, San Diego, USA), for the assessment of anti-topoisomerase I (anti-Scl70) antibodies Orgetec, ORG 514, Mainz, Germnay ELISA assay was used. Anticentromer antibodies were tested with Orgetec, ORG 633, Mainz, Germany ELISA assay and anti-RNA polymerase III was evaluated with Euroimmune DL 15321601, Mountain Lake, USA immunoblot assay. Based on literature ANA negative patients are prone to have less severe vascular abnormalities (including lower number of teleagiectasia, digital uclers) [doi: 10.1111/1756-185X.13908. and doi: 10.1016/j.semarthrit.2014.11.006.]. In our cohort patients with SSc positive for ANA had significantly higher CD compared to ones without ANA positivity (6.0/4.9;7.5/ vs. 7.8/7.0;8.4/ p<0.001), and prevalence of SSc pattern and elevated MES was also significantly elevated in these particular patients (88.5% vs. 46.4%, p<0.001 and 86.9% vs. 60.7%; p<0.001, respectively). These results are in line with previous publications, but the high prevalence of ANA negative patients might made influence on the results. The low prevalence of ANA positivity might be explained by selection bias or method of assessment of ANA.
2. All CTD patients were included in the analysis. What were the CD and SSc-pattern data when only patients with Raynaud's phenomenon were considered?
Patients with RP in case of SLE and RA had significantly decreased CD compared to their counterparts without RP. No significant difference on SSc pattern was observed on comparison of patients with and without RP in patients with CTD patients. Please, see details in the revised form of our article.
3. Characteristics of "SSc overlap" and "CTD overlap" groups are missing
Characteristics of SSc of overlap patients were already included in our Table 1, but based on your recommendation we have added the clinical features of overlap CTD patients also. Thank you.
4. The conclusions are too general
We tried to better emphasize the main observations of our study.
Hopefully our answers achieve your standards and You will find our manuscript valuable for publication.
Yours sincerely.
Gábor Kumánovics MD, PhD
Reviewer 3 Report
The study presented by Nagy et al is well written and presented. I have only some minor comments:
- Did you asses laboratory parameters in your patients (Creatinine, CRP, etc.)? Are there associations?
- Which other comorbidities had the patients? Could you provide a multiple regression analysis for those?
- Did you also perform other measurements, e.g. sublingual capillary density assessment, thermography, etc.?
- Capillary density rarefaction and other microvascular alterations are observed widely across different other pathologies (heart failure, diabetes, chronic kidney disease, acquired von Willebrand disease). Please discuss in relation to your paper.
Author Response
Dear Reviewer III,
Thank for reviewing our manuscript. Regarding your comments please find our answers below.
- Did you asses laboratory parameters in your patients (Creatinine, CRP, etc.)? Are there associations?
Laboratory parameters (e.g. creatinine or CRP) were not assessed in our cohort therefore evaluation of the possible connection to different capillary parameters is not possible.
2. Which other comorbidities had the patients? Could you provide a multiple regression analysis for those?
Only presence of hypertension and diabetes mellitus was recorded in our database at data collection. There are very low number of patients in different subgroups, therefore multiple regression analysis might lead to invalid results.
3.Did you also perform other measurements, e.g. sublingual capillary density assessment, thermography, etc.?
Beside assessment of capillary density, SSc pattern and MES other semi quantitative parameters introduced by Sulli (doi: DOI: 10.1136/ard.2007.079756.) were investigated in our cohort including giant capillary, haemorrhage and dilatation score. Data on these parameters (as they are only validated in systemic sclerosis and mainly used for scientific purposes) were not presented in our manuscript, we aimed to focus on more accepted and conventional parameters. Capillary measurement were only performed on fingers, other possible are (for example lips or toes) were not included in our study design.
- Capillary density rarefaction and other microvascular alterations are observed widely across different other pathologies (heart failure, diabetes, chronic kidney disease, acquired von Willebrand disease). Please discuss in relation to your paper.
Limitation of our study might be that different conditions including hypertension or diabetes mellitus, might be characterized by capillary rarefaction (doi: 10.2152/jmi.68.6. and doi: 10.1016/j.mvr.2020.104036.) were not evaluated in our study.
Hopefully our answers achieve your standards and You will find our manuscript valuable for publication.
Yours sincerely.
Gábor Kumánovics MD, PhD
Round 2
Reviewer 1 Report
I am happy to see that the authors improved the manuscript.
Further aspects to consider:
- Please add power analyses to the manuscript.
- Analyse the impact of treatment, or at least acknowledge this as a limitation.
Author Response
Dear Reviewer,
Thank you for reviewing our manuscript. Regarding your comments, please find our response below.
- Please add power analyses to the manuscript.
A priori sample size calculation was based on capillary density, to best differentiate between different CTDs and healthy volunteers.
The manuscript was corrected as follows:
Based on previous publications an a priori sample size calculation with a power of 80% and α = 0.05 was performed to best differentiate CD between HVs and patients with different CTDs [28,41-44].
2. Analyse the impact of treatment, or at least acknowledge this as a limitation.
Besides activity, treatment may influence capillary pattern and density also. It was described in myositis that capillary pattern may change rapidly during disease course and treatment, furthermore short series were published on patients with systemic sclerosis that immunosuppressive treatment may improve capillary density. Patient selection possibly also influenced our results as both treatment naïve and treated patients were enrolled in the study. To evaluate the possible effect of treatment follow up studies are suitable instead of our cross-sectional study design. We added this remark to the limitations of our study.
Hopefully, our answers achieve your standards and You will find our manuscript valuable for publication.
Yours sincerely,
Gábor Kumánovics MD, PhD
Reviewer 2 Report
The authors revised the article as recommended by the reviewers. Only a few minor issues remain:
- please provide proper reference citations i.e. [9-15] instead of "[9][10][11][12][13][14][15]"
- please give short description in Results section of types of overlap with systemic sclerosis (scleromyositis? secondary Sjögren syndrom?)
Author Response
Dear Reviewer,
Thank you for reviewing our manuscript. Regarding your comments, please find our response below.
- Please provide proper reference citations i.e. [9-15] instead of "[9][10][11][12][13][14][15]"
Thank you for your suggestion. We corrected the references based on your recommendation.
- please give short description in Results section of types of overlap with systemic sclerosis (scleromyositis? secondary Sjögren syndrom?)
Overlap SSc was detected in 39 cases, among them 29 patients fulfilled two classification criteria. SSc-SLE overlap was present in 6 cases, SSc-APS overlap was detected in 2 cases, SSc-IIM in 5 cases, SSc-SS overlap in 9 cases, RA overlap in 6 cases and one patient had ANCA associated vasculitis beside SSc. In case of 8 patients 3 different classification criteria was fulfilled (SSc-SLE-APS n=1, SSc-SLE-SS n=3, SSc-SS-RA n=2, SSc-SLE-RA n=1 and SSc-IIM-SS n=1, respectively). In case of 2 patients SSc-SLE-SS-RA was classified. The high prevalence of the multiple overlaps might be explained by that we are a tertiary care center and we follow up a substantial number of complex cases of the region.
Hopefully, our answers achieve your standards and You will find our manuscript valuable for publication.
Yours sincerely.
Gábor Kumánovics MD, PhD